# Do ESG Ratings of Chinese Firms Converge or Diverge? A Comparative Analysis Based on Multiple Domestic and International Ratings

**Yunfu Zhu** **, Haoling Yang and Ma Zhong \***

College of Economics and Management, Nanjing Forestry University, No. 159 Longpan Road, Xuanwu District, Nanjing 210037, China; zhuyunfu@njfu.edu.cn (Y.Z.); yhl@njfu.edu.cn (H.Y.)
\* Correspondence: zm6040@njfu.edu.cn

**Abstract:** Since the Chinese economy has transitioned to a sustainable model, the Chinese socially responsible investment (SRI) market has expanded rapidly, which has deeply stimulated the development of environmental, social, and governance (ESG) ratings for Chinese firms. Domestic agencies, such as SynTao, Rankins (RKS), Sino-Securities (SSII), and China Alliance of Social Value Investment (CASVI), and international agencies, such as Bloomberg, FTSE Russell (FTSE), and Morgan Stanley Capital International (MSCI), have launched their own ESG rating systems. These emerging ratings may provide users of information with more diverse references; however, if their results are too divergent, they may also confuse users. To what extent do these ESG rating results in the Chinese market converge or diverge? Aiming to answer this question, we used Hushen 300 index firms in 2019 as the initial sample, and selected 195 firms covered by the above seven ratings for the analysis. Firstly, by comparing the overlap in the top 100 lists of these sample firms, we found that the list overlap rate between each pair of ratings was between 66.36% and 82.35%; however, only 35% of the firms were listed in the top 100 of all seven ratings. Furthermore, the Pearson correlation analysis showed that the correlation coefficients between each pair of ratings ranged from 0.057 to 0.736, and the average was only 0.411. These results suggest a wide divergence in the ESG rating results for Chinese firms. We suggest that information users need to consider a more diverse and comprehensive perspective when utilizing these ratings.

**Keywords:** ESG ratings; emerging markets; rating divergence; corporate sustainability



## 1. Introduction

After nearly two decades of rapid growth, China has become the world's second-largest economy, and the largest developing economy. However, given this radical economic development, various environmental and social scandals have continuously broken out, e.g., the Sanlu milk scandal in 2008 [1] and the Zijin Mining scandal in 2010 [2], and the problems in Chinese society that are associated with sustainable development have become more prominent [3]. In this regard, the sustainable transformation of economic growth was necessary, and the Chinese government has made many attempts, such as promulgating a new environmental law in 2015, and so on. Spurred by sustainable transformation, the Chinese socially responsible investment (SRI) market has grown rapidly. According to Syntao [4], the total size of the Chinese SRI market reached approximately RMB 24.6 trillion in 2021. The SRI market drives the development of ESG ratings for Chinese firms. For example, local sustainability consultancies SynTao and Rankins (RKS) launched their respective ESG rating services in 2015 and 2019, respectively; international agencies Bloomberg, FTSE Russell (FTSE), and Morgan Stanley Capital International (MSCI) are also expanding their coverage of Chinese listed firms (e.g., FTSE ESG's coverage of Chinese firms was only 30 in 2018, but reached 679 in 2019). The ratings issued by different agencies bring many references and information to users, but may cause confusion if their

results show excessive divergence. Using recent academic research as an example, among the 41 pieces of literature collected in this paper on the topic of Chinese ESG, most only use two or fewer ratings (92.68%) as a proxy for ESG performance or disclosure, and 53.66% use only one rating. If ESG ratings diverge too much, the reliability and robustness of the conclusions of these studies may be in doubt. Therefore, we raise the following research questions: do significant divergences exist? To what extent do these ESG rating results converge or diverge?

In view of the above questions, we used 195 Hushen 300 index firms commonly covered by seven domestic and international ESG ratings in 2019 as samples to evaluate their convergence (the above seven ESG ratings are considered by some studies to have an important impact on the Chinese market, e.g., Li, Yin, and Liu (2022) [5] and Liu (2022) [6]). First, we investigated the overlap between the top 100 lists of these sample firms covered by all seven ratings, and the results showed that the overlap rates are not high, e.g., the overlap rate among all ratings is only 35%. Then, the correlation analysis showed similar results, e.g., that the correlation coefficient between the pairwise ratings is between 0.057 and 0.736, and the average coefficient is only 0.411. The above results suggest that these ratings of Chinese firms show high divergence (low convergence). Therefore, we suggest that researchers and investors need to compare and synthesize these ratings. The contributions of this paper are mainly that it enriches the research on the validity of ESG ratings [7–14], supplements the recognition of ESG rating divergence (convergence) in the Chinese market, and helps investors and researchers interested in the Chinese market to more intuitively understand the various ESG rating results in the Chinese market.

This study has five sections. Section 2 presents the background and develops a rating system comparison on ESG ratings in the Chinese market. Section 3 conducts a convergence analysis of these ESG ratings. Section 4 contains the discussion, and Section 5 draws the conclusions.

## 2. Background and Comparison of ESG Ratings in China
### 2.1. Background of ESG Ratings in China
2.1.1. Advent and Development of ESG Ratings in China

Table 1 summarizes the development of domestic and international ESG ratings for Chinese firms. As an established sustainability consultancy in China, SynTao was the first domestic agency to carry out an ESG rating business in 2015. Its initial rating range only covered Hushen 300 firms, and then was expanded yearly until 2022 to cover all A-share listed firms. In 2017, the financial index firm SSII and nongovernmental organization CASVI also started to provide ESG ratings. However, the coverage of the two ratings varies widely, with SSII covering all A-share listed firms, and CASVI only covering the Hushen 300. In 2020, corporate social responsibility (CSR) consultancy RKS ceased its original CSR ratings business, and transformed to provide ESG rating services covering CSI 800 index firms. Bloomberg offered the first international ESG rating to cover firms in the Chinese market. Its database shows that its earliest ESG rating year involving a Chinese A-share firm is 2011. MSCI's involvement came after it included Chinese A-share firms in its emerging market index and global index in 2018, and began offering ESG ratings for Chinese A-share firms in 2019. Similarly, FTSE Russell began assigning ESG ratings to Chinese A-share firms after covering them in its global index in 2019.

2.1.2. Application of ESG Ratings in China

Investigating the application of ESG ratings by market practitioners is limited by objective factors. Therefore, we collected 41 studies from the last five years that address or are related to ESG issues in the Chinese market, and that each study included the determinants or economic consequences of ESG performance or disclosure (ESG ratings are used in their research design; a brief description of the literature collection is reported in Appendix A). We counted the application of seven ESG ratings, as described below.

**Table 1.** Basic information on ESG ratings.

| Rating | Category | Start Time | Attributes | Coverage |
|---|---|---|---|---|
| Bloomberg | International | 2006 | Index/data provider | Approximately 860 A-share firms in 2011, over 1200 after 2020 |
| FTSE | International | 2019 | Index/data provider | Covering 30 in 2018, rising to 679 in 2019, and approximately 900 A-share listed firms by the end of 2022 |
| MSCI | International | 2018 | Index/data provider | Covering 5% of A-share listed firms in 2018, raised to 20% in 2019 |
| SynTao | Domestic | 2015 | Consultancy | Hushen 300 (2015), CSI 500 (from 2018), now covering all A-share listed firms |
| RKS | Domestic | 2020 | Consultancy | CSI 800 |
| SSII | Domestic | 2017 | Index/data provider | All A-share listed firms |
| CASVI | Domestic | 2017 | NGO | Hushen 300 |

Firstly, regarding the agency category, in all 41 studies, domestic ESG ratings were used 35 times (85.37%), and international ESG ratings were used 25 times (60.98%) (Since some studies used two or more ratings simultaneously, the sums of the relevant proportions calculated in this paper are not equal to 100%. For example, when Tian (2022) [15] studied the impact of ESG performance on corporate trade credit financing, SSII ESG was used as a proxy in the main analysis. Bloomberg ESG was used for robustness). Concretely, as shown in Figure 1, in the literature using domestic ratings, the frequencies (rates) of SSII, SynTao, CASVI, and RKS were 20 (48.78%), 8 (19.51%), 4 (9.76%), and 3 (7.32%), respectively. Among them, SSII is the most frequently used, which may be due to its widest coverage (covering all A-share listed firms). In the literature using international ratings, the frequencies (rates) of Bloomberg, MSCI, and FTSE were 18 (43.90%), 5 (12.20%), and 2 (4.88%), respectively. Perhaps because Bloomberg covers Chinese firms at the earliest times and has a larger reach, it is the most used. Among all of the ratings, the top three most-used were SSII (48.78%), Bloomberg (43.90%), and SynTao (19.51%).

Secondly, as shown in Figure 2, the number of studies using two or fewer ratings in one study was the largest (38 papers, accounting for 92.68%); 53.66% of the studies (22 papers) used only one rating; and only 7.32% (3 papers) used three or more ratings. If there is a large divergence between different ratings, the conclusions of these studies may suffer from robustness issues. For developed capital markets, scholars carried out some studies on ESG divergence and came to similar conclusions to those we advocate, that this robustness issue is widespread [7,9,16].

### 2.2. Comparison of Various ESG Ratings

We made a comparison from three aspects: metric system, information source and operating process:

(1) Metric system. As shown in Panel A of Table 2, all of the six ratings but CASVI have environment, social, and governance as level-1 dimensions. There is a wide variation in the number of underlying metrics for each rating, with the fewest indicators being SSII (including 26 indicators) and the most being FTSE (including over 300 indicators). Thus, we present a comparative analysis using the related topics of the indicators as classification dimensions, and the results are reported in Table 2 Panel B. Regarding the environmental dimension, all seven ratings involved "Climate change", "Sewage discharge", "Environmental supply chain", "Exhaust gas emission", and "Solid waste discharge". Furthermore, "Biodiversity", "Water resource management", "Negative environmental events", and "Green purchasing" received attention from each rating, except for CAVSI. The social dimension varied greatly, and the issues of most concern were "Health and safety", "Supply chain management", "Employee compensation", and "Product safety". However, international rating agencies pay more attention to "Human rights", while Chinese domestic agencies pay more attention to "Corporate donation", "Information security", and "Inclusive finance". Regarding the

governance dimension, the issues with the most concerns were "Audit and supervision", "Board structure", "Independent directors", "Fighting corruption", and "Business ethics". Specifically, domestic institutions (such as SSII and CASVI) pay attention to creditor rights and interests. In the Chinese economy, debt financing plays a dominant role. The balance of RMB loans in the real economy at the end of 2019 accounted for 60.3% of the social financing stock during the same period, according to the 2019 Social Financing Stock Statistics Report. The balance of entrusted loans was 4.6%; the balance of trust loans was 3%; the balance of undiscounted bank acceptance bills was 1.3%; the balance of corporate bonds was 9.3%; the balance of government debt was 15%; and the balance of domestic stocks of nonfinancial firms was 2.9%. Finally, each rating has different practices in determining the indicator weights. For example, MSCI sets weights based on the length and magnitude of the expected impact of an ESG issue; FTSE calculates weights based on the exposure of issues; and RKS ratings determine weights based on investor attention to the issue.

**Table 2.** Comparison of ESG rating metric systems.

**Panel A**

| Rating Agency | Framework Level | Level-1 Division | Number of Underlying Metrics |
|---|---|---|---|
| Bloomberg | 3 | Environment, social, and corporate governance | 61 |
| FTSE | 3 | Environment, social, and corporate governance | 300+ |
| MSCI | 3 | Environment, social, and corporate governance | 37 |
| SynTao | 3 | Environment, social, and corporate governance | 200+ |
| RKS | 3 | Environment, social, and corporate governance | 37 |
| SSII | 3 | Environment, social, and corporate governance | 26 |
| CASVI | 4 | Objective (driving force), approach (innovation), effectiveness (implementation) | 55 |

**Panel B**

| Dimension | Issue | Bloomberg | FTSE | MSCI | SynTao | RKS | SSII | CASVI | Frequency |
|---|---|---|---|---|---|---|---|---|---|
| Environment | Solid waste discharge | √ | √ | √ | √ | √ | √ | √ | 7 |
| | Climate change | √ | √ | √ | √ | √ | √ | √ | 7 |
| | Sewage discharge | √ | √ | √ | √ | √ | √ | √ | 7 |
| | Environmental supply chain | √ | √ | √ | √ | √ | √ | √ | 7 |
| | Exhaust gas emission | √ | √ | √ | √ | √ | √ | √ | 7 |
| | Biodiversity | √ | √ | √ | √ | √ | √ | — | 6 |
| | Water resource management | √ | √ | √ | √ | — | √ | √ | 6 |
| | Negative environmental events | √ | √ | — | √ | √ | √ | √ | 6 |
| | Green purchasing | √ | √ | √ | √ | — | √ | √ | 6 |
| | Energy issues | √ | √ | — | √ | — | √ | √ | 5 |
| | Green products | — | — | √ | √ | √ | √ | — | 4 |
| | Renewable energy management | √ | — | √ | √ | — | √ | — | 4 |
| | Packaging materials | — | — | √ | — | √ | — | — | 2 |
| Society | Health and safety | √ | √ | √ | √ | √ | √ | √ | 7 |
| | Supply chain management | √ | √ | √ | √ | √ | √ | √ | 7 |
| | Product safety | √ | √ | √ | √ | √ | √ | √ | 7 |
| | Employee compensation | √ | √ | √ | √ | √ | √ | √ | 7 |
| | Staff training | √ | — | √ | √ | √ | √ | √ | 6 |
| | Negative events | — | √ | √ | — | √ | √ | √ | 5 |
| | Communities | √ | — | — | √ | √ | √ | √ | 5 |
| | Customer management | √ | √ | — | √ | — | √ | √ | 5 |
| | Corporate donation | — | — | — | √ | √ | √ | √ | 4 |
| | Employee diversity | √ | — | — | √ | √ | — | √ | 4 |
| | Information security | — | — | √ | √ | √ | √ | — | 4 |
| | Human rights | √ | √ | √ | √ | — | — | — | 4 |
| | Inclusive finance | — | — | — | √ | √ | √ | — | 4 |
| Governance | Fighting corruption | √ | √ | √ | √ | √ | √ | √ | 7 |
| | Board structure | √ | √ | √ | √ | √ | √ | √ | 7 |
| | Independent directors | √ | √ | √ | √ | √ | √ | √ | 7 |
| | Audit and supervision | √ | √ | √ | √ | √ | — | √ | 6 |
| | Business ethics | √ | — | √ | √ | √ | √ | √ | 6 |
| | Executive compensation | √ | — | √ | √ | √ | — | — | 4 |
| | Risk management | — | √ | — | — | √ | √ | √ | 4 |
| | Tax transparency | — | √ | √ | √ | — | √ | — | 4 |
| | ESG disclosure | — | √ | — | √ | — | √ | √ | 4 |
| | Debt-paying ability | — | — | — | — | — | √ | √ | 2 |

Note: the data are collected manually. A "√" indicates that the issue is involved in this rating, and a "—" indicates that it is not involved. The "Frequency" column represents the number of ratings that focus on the issue.

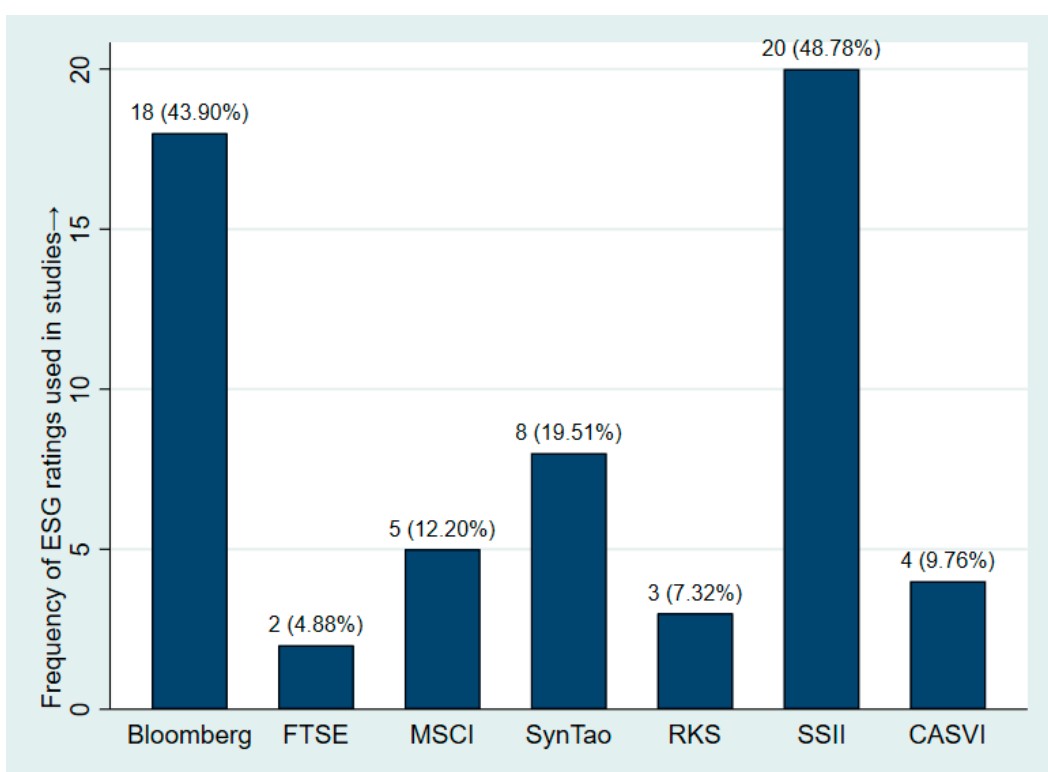

**Figure 1.** Frequency of use of ESG ratings in recent years in Chinese ESG-related studies. Note: rate of use is in parentheses.

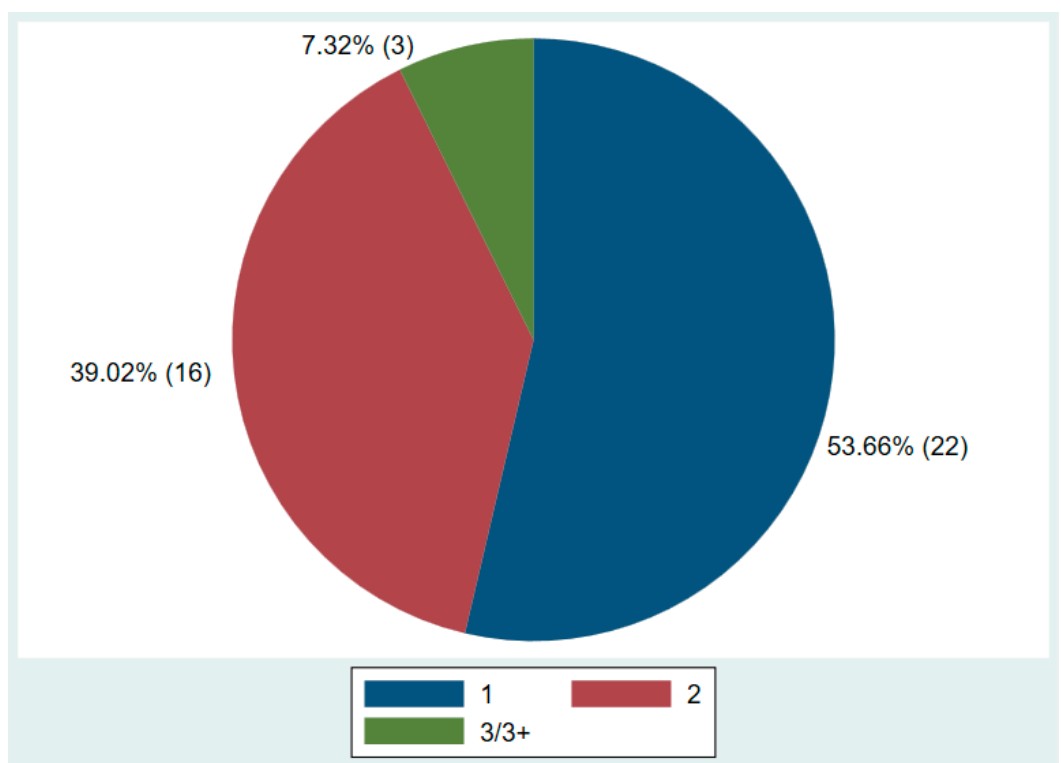

**Figure 2.** Numbers of ESG ratings used in China ESG studies in recent years. Note: "1" and "2" indicate that the paper uses only one and two types of ESG ratings, and "3/3+" indicates that the paper uses three or more ESG ratings. Note: number of papers is in parentheses.

(2) Information source. A comparison of the information sources is reported in Table 3. Among them, the information sources of each rating are based on the firm's public announcements, mainly including annual and CSR/ESG reports; however, there are differences in other supplementary information. Ratings such as MSCI and SynTao include corporate social media disclosures in their sources; FTSE uses a more proactive strategy, including communicating with the rated firms to investigate their undisclosed information. However, none of the ratings mention the weights (or a similar description) of different sources in their ratings.

**Table 3.** Comparison of information sources for ESG ratings.

| Rating Agency | Information Sources Claimed by the Rating Agency |
|---|---|
| Bloomberg | Public information such as the firm's sustainability reports, financial statements, website, and press releases. |
| FTSE | Mainly from public information from listed firms, regulators, news media, and other NGOs. |
| MSCI | Public data: government, regulatory, and NGO datasets; firm disclosure documents; 3400 media sources. |
| SynTao | The positive information mainly comes from the independent disclosure of firms, while the negative information comes from the independent disclosure of firms, media reports, announcements of regulatory authorities, and surveys of social organizations. |
| RKS | Mainly comes from the information independently disclosed by the firm, including the firm's annual report, corporate social responsibility report, and articles of association. |
| SSII | Open data based on a machine learning and text mining algorithm that crawls the website data of the government and relevant regulatory authorities, news media data, etc. |
| CASVI | Public information, including the firm's annual report, social responsibility report, government departments and third-party announcements, etc. |

(3) Operating process. All ratings basically follow these four processes: a. gather raw information; b. sort the raw information and fill in the underlying metrics; c. summarize the underlying metrics to form the rating score; and d. review the scores to form the final result. However, the specific process details of ratings differ. A review process is used as an example (as shown in Table 4): Bloomberg and SynTao adopt only internal reviews, while other agencies adopt external reviews or joint internal and external reviews. In addition, none of the ratings are cross-reviewed or communicated to other peers (e.g., other ESG rating agencies). According to the illustration released by the ratings, there was no cross-review or communication with peers for not only the review process, but also for other processes.

**Table 4.** Comparison of review processes for ESG ratings.

| Rating Agency | Review Process |
|---|---|
| Bloomberg | Review the scoring framework internally and establish communication channels with the firm. |
| FTSE | Oversight by external committees set up by businesses, investors, nonprofit organizations, experts, and academics. |
| MSCI | Invite the evaluated firms to review relevant information and data before issuing the final score. |
| SynTao | Conduct cross audits within the organization. |
| RKS | A technical committee composed of experts and scholars from external universities and research institutions can review the rating results. |
| SSII | An ESG expert committee is set up to review the rating results. |
| CASVI | Hire data experts to review and verify information with regulatory authorities and industry associations. |

The above comparative analysis results show that each ESG rating has its own preference in terms of its metric system, information source, and process design, and there is a lack of a cross-review or communication mechanism among the ratings. This situation intensifies our concern over the convergence (divergence) problem among the rating results.

### 3. Analysis for the Convergence/Divergence of Chinese ESG Ratings

*3.1. Analysis Design*

In this section, we used the 2019 Hushen 300 index firms as the initial sample and filtered out 195 sample firms that are jointly covered by all seven agencies. Referring to the historical studies [7,9,11,14,16], we used ranking overlap analysis and correlation analysis to intuitively show the convergence/divergence between these ratings. First, we investigated the overlap of these sample firms ranked in the top 100 list of each rating (We also attempted to analyze the extent of overlap between the top 50 and top 30 lists and finds similar results. For brevity, this will not be repeated, and related results are available from the authors on request). Then, we conducted correlation analysis on the rating scores of these sample firms.

*3.2. Sampling Process*

The original sample was the firms in the Hushen 300 Index of 2019 for the following reasons. First, the firms in the Hushen 300 index are sufficiently representative in terms of coverage. In 2019, the total market value of the firms in the Hushen 300 index accounted for 80.96% of all A-share listed firms in China (calculated using market capitalization data in the CSMAR database), and most were also included in various international indices (e.g., the MSCI Global Index). Second, since the outbreak of COVID-19 in 2020, strict Chinese prevention and control measures have severely affected various corporate activities, including ESG activities (Regarding a direct impact, COVID-19 has affected the focus of corporate ESG activities. For example, many firms have shifted their focus to social issues, such as anti-epidemic donations. Regarding indirect impacts, strict epidemic control has restricted business activities and has reduced firms' profitability, forcing some to scale back substantial investments in sustainable responsibility). To avoid these effects, we chose 2019—before the outbreak of the pandemic—as the sample period in this paper.

The screening process of the sample is reported in Table 5. Among the ratings, SSII has the largest coverage of all 3775 A-share listed firms, while CASVI has the smallest coverage of 410 firms. Among the international ratings, Bloomberg has the largest coverage (1207), and MSCI has the smallest coverage (580). For the Hushen 300 firms selected in this paper, the number of firms covered by domestic ratings ranges from 268 (CASVI) to 300 (SSII), and the international ratings cover from 250 (Bloomberg) to 283 (MSCI). In the end, 195 Hushen 300 firms at all seven levels of institutions were fully covered.

**Table 5.** Samples selection.

| Rating Agency | Total Number of Chinese Firms Covered in 2019 | Number of Hushen 300 Firms Covered in 2019 | Number of Hushen 300 Firms Commonly Covered by All Ratings in 2019 |
|---|---|---|---|
| Bloomberg | 1207 | 250 | 195 |
| FTSE | 679 | 255 | 195 |
| MSCI | 580 | 283 | 195 |
| SynTao | 1267 | 279 | 195 |
| RKS | 790 | 294 | 195 |
| SSII | 3775 | 300 | 195 |
| CASVI | 410 | 268 | 195 |

*3.3. Descriptive Statistics*

Table 6 reports the descriptive statistics, including observation, mean, standard deviation (SD), median, and so on, of ESG rating scores based on the 195 sample firms. Panel A shows the statistical results before standardization, in which Bloomberg, FTSE, MSCI, SynTao, and RKS publish continuous scores, with means (medians) of 30.55 (28.93), 1.59 (1.6), 37.71 (35.16), 52.12 (51.13), and 2.97 (2.69), respectively. The results published by SSII and CASVI are graded scores that are converted into corresponding scores according to their grades in this paper (The SSII rating contains nine grades: AAA, AA, A, BBB, BB, B,

CCC, CC, and C. Therefore, we transformed this rating into 0 to 9. The CASVI rating contains 20 grades (from D to AAA), and we transformed it into 0 to 20), with means (medians) of 5.03 (4) and 11.97 (12), respectively. To further directly compare the differences among the results of the ratings, we reported the statistics in panel B after 0–1 standardization. The scoring results of Bloomberg, FTSE, MSCI, SynTao, and RKS are more conservative, with the mean (median) ranging from 0.354 (0.353) to 0.452 (0.458), with all being less than 0.5. Meanwhile, the results of SSII and CASVI are more optimistic, with means (medians) of 0.605 (0.6) and 0.61 (0.611), respectively, both being higher than 0.5. The above results indicate that the scoring results of different ratings have different tendencies: relatively, the ratings of Chinese firms by international agencies are more conservative; among domestic ratings, established sustainability consultants SynTao and RKS also report more conservative results, but the financial index firm SSII and nongovernmental organization CASVI are more optimistic.

**Table 6.** Descriptive statistics of ESG rating scores.

| **Panel A (Before 0–1 Standardization)** | | | | | | | | |
|---|---|---|---|---|---|---|---|---|
| **Rating Agency** | **N** | **Mean** | **SD** | **Min** | **P25** | **P50** | **P75** | **Max** | **Scoring Range** |
| Bloomberg | 195 | 30.55 | 10.52 | 11.98 | 21.49 | 28.93 | 39.26 | 59.92 | 0.1–99 |
| FTSE | 195 | 1.59 | 0.54 | 0.50 | 1.20 | 1.60 | 2.00 | 2.90 | 0–5 |
| MSCI | 195 | 37.71 | 16.29 | 3.17 | 26.53 | 35.16 | 49.58 | 81.02 | 0–100 |
| SynTao | 195 | 52.12 | 6.39 | 39.13 | 47.63 | 51.13 | 56.75 | 71.88 | 0–100 |
| RKS | 195 | 2.97 | 1.45 | 0.76 | 1.85 | 2.69 | 3.84 | 7.00 | 0–7 |
| SSII | 195 | 5.03 | 1.11 | 2.00 | 2.00 | 4.00 | 5.00 | 6.00 | 0–9 |
| CASVI | 195 | 11.97 | 4.01 | 1.00 | 10.00 | 12.00 | 15.00 | 19.00 | 0–20 |
| **Panel B (After 0–1 Standardization)** | | | | | | | | |
| **Rating Agency** | **N** | **Mean** | **SD** | **Min** | **P25** | **P50** | **P75** | **Max** | **Scoring Range** |
| Bloomberg | 195 | 0.387 | 0.220 | 0.017 | 0.198 | 0.353 | 0.569 | 0.879 | 0–1 |
| FTSE | 195 | 0.452 | 0.226 | 0.042 | 0.292 | 0.458 | 0.625 | 0.958 | 0–1 |
| MSCI | 195 | 0.444 | 0.209 | 0.005 | 0.300 | 0.411 | 0.596 | 0.952 | 0–1 |
| SynTao | 195 | 0.397 | 0.195 | 0.027 | 0.260 | 0.366 | 0.538 | 0.905 | 0–1 |
| RKS | 195 | 0.354 | 0.232 | 0.010 | 0.175 | 0.309 | 0.494 | 0.952 | 0–1 |
| SSII | 195 | 0.605 | 0.221 | 0.000 | 0.400 | 0.600 | 0.800 | 1.000 | 0–1 |
| CASVI | 195 | 0.610 | 0.223 | 0.000 | 0.500 | 0.611 | 0.778 | 0.944 | 0–1 |

*3.4. Ranking Overlap Analysis*

Table 7 shows the results of the analysis of the overlap based on the top 100 lists for each rating among the 195 sample firms, where panels A and B represent the results of the overlap frequency and rate of the rankings, respectively (When there was a difference in the number of top 100 lists between two ratings, we calculated the overlap rate using the smaller number as the denominator. For example, the top 100 list of the SSII contains 133 firms, the CASVI contains 119, and the number of firms covered by the two together is 98, so the overlap rate is 82.35% = (98/119) × 100%). The diagonal line in panel A shows the number of the top 100 sample firms in each rating. Due to the use of the grade rating system, the top 100 of SSII and CASVI contain 133 and 119 sample firms, respectively. The frequency of the top 100 firms that overlap each rating are in each column. For example, column (1) of panel A and column (1) of panel B show the frequency and rate of overlapping firms between Bloomberg and other ratings, respectively (As noted above, we used the lesser of the two ratings as the denominator, and the overlap rate (between the two ratings) = (84/103) × 100% = 81.5%); the lowest is Bloomberg with CASVI, with an overlap frequency (rate) of 75 (72.82%). For the statistics of the pairwise overlap of all ratings, the lowest overlap occurs between FTSE and CASVI, with a frequency (rate) of overlap of 71 (66.36%). The highest overlap frequency (rate) is 98 (82.35%). Finally, we reported the overlap of international agencies, domestic agencies, and all agencies: the

overlap frequency (rate) of the top 100 lists of three international ratings (Bloomberg, FTSE, and MSCI) is 57 (57% = 57/100); the frequency (rate) of overlap among the four domestic ratings (SynTao, RKS, SSII, and CASVI) is 46 (46%); and the frequency (rate) of overlap among all seven ratings is only 35 (35%), which means that only approximately one-third of the sample firms are jointly recognized by the seven ratings in the top 100.

**Table 7.** Overlap of top 100 firms of different ratings.

| Panel A (Overlap Frequency of the Top 100 Sample Firms) | | | | | | | |
|---|---|---|---|---|---|---|---|
| | **Bloomberg** | **FTSE** | **MSCI** | **SynTao** | **RKS** | **SSII** | **CASVI** |
| (1) Bloomberg | 103 | | | | | | |
| (2) FTSE | 84 | 107 | | | | | |
| (3) MSCI | 77 | 78 | 100 | | | | |
| (4) SynTao | 81 | 78 | 72 | 100 | | | |
| (5) RKS | 82 | 77 | 75 | 74 | 100 | | |
| (6) SSII | 81 | 82 | 77 | 78 | 77 | 133 | |
| (7) CASVI | 75 | 71 | 71 | 69 | 70 | 98 | 119 |
| (8) | | | | Overlap frequency of international ratings: 57 | | | |
| (9) | | | | Overlap frequency of domestic ratings: 46 | | | |
| (10) | | | | Overlap frequency of all ratings: 35 | | | |
| **Panel B (Overlap Rate of the Top 100 Sample Firms)** | | | | | | | |
| | **Bloomberg** | **FTSE** | **MSCI** | **SynTao** | **RKS** | **SSII** | **CASVI** |
| (1) Bloomberg | 100% | | | | | | |
| (2) FTSE | 81.55% | 100% | | | | | |
| (3) MSCI | 77.00% | 78.00% | 100% | | | | |
| (4) SynTao | 81.00% | 78.00% | 72.00% | 100% | | | |
| (5) RKS | 82.00% | 77.00% | 75.00% | 74.00% | 100% | | |
| (6) SSII | 78.64% | 76.64% | 77.00% | 78.00% | 77.00% | 100% | |
| (7) CASVI | 72.82% | 66.36% | 71.00% | 69.00% | 70.00% | 82.35% | 100% |
| (8) | | | | Overlap rate of international ratings: 57% | | | |
| (9) | | | | Overlap rate of domestic ratings: 46% | | | |
| (10) | | | | Overlap rate of all ratings: 35% | | | |

*3.5. Correlation Analysis*

Table 8 shows the results of the Pearson correlation analysis of the ESG rating scores for the 195 sample firms (The Spearman correlation analysis results are similar and reported in Appendix B). Pearson correlation analysis is commonly used to recognize the linear correlation between two variables [17,18]. In general, the correlation coefficients between the two ratings are extremely variable, ranging from 0.057 (FTSE and CASVI) to 0.736 (Bloomberg and SynTao), with an average of 0.411. Since the results released by SSII and CASVI are grade ratings, they also differ from the other ratings. When they are excluded, the correlation coefficient between pairwise ratings ranges from 0.550 (RKS and SynTao) to 0.736 (Bloomberg and SynTao), with a mean of 0.622; however, approximately half of the ratings still do not exceed 0.6. Based on each rating, Bloomberg has the highest correlation with other ratings, with an average correlation coefficient of 0.540 (0.658 when SSII and CASVI are not considered); the other results (FTSE, MSCI, SynTao, RKS) are also similar, with mean correlation coefficients ranging from approximately 0.4 to 0.5 (approximately 0.6 when SSII and CASVI are not considered). Due to the use of ratings, SSII and CASVI have lower correlation coefficients with other ratings, with mean coefficients of 0.251 and 0.223, respectively; however, the correlation between the two is only 0.427. The above results and the previous ranking overlap analysis also support our concerns. Due to differences in the rating systems and methodologies, the correlations of individual ratings are not strong, and the correlations between pairwise ratings are mostly below 0.7. Even after excluding SSII and CASVI, which use the grade ranking approach, the average correlation coefficient

for the other five ratings is only approximately 0.6, indicating a divergence between these rating results.

**Table 8.** Pearson correlations between ESG ratings.

|  | Bloomberg | FTSE | MSCI | SynTao | RKS | SSII | CASVI |
|---|---|---|---|---|---|---|---|
| Bloomberg | 1 |  |  |  |  |  |  |
| FTSE | 0.591 *** | 1 |  |  |  |  |  |
| MSCI | 0.682 *** | 0.566 *** | 1 |  |  |  |  |
| SynTao | 0.736 *** | 0.607 *** | 0.587 *** | 1 |  |  |  |
| RKS | 0.621 *** | 0.686 *** | 0.597 *** | 0.550 *** | 1 |  |  |
| SSII | 0.310 *** | 0.114 | 0.223 *** | 0.274 *** | 0.155 ** | 1 |  |
| CASVI | 0.300 *** | 0.057 | 0.261 *** | 0.171 ** | 0.121 * | 0.427 *** | 1 |
| Mean of coefficients | 0.540 (0.658) | 0.437 (0.613) | 0.486 (0.608) | 0.488 (0.620) | 0.455 (0.614) | 0.251 | 0.223 |

Note: * $p < 0.1$, ** $p < 0.05$, *** $p < 0.01$; the values in parentheses are the coefficient means without considering SSII and CASVI.

## 4. Discussion

The evidence in this paper indicates that ESG ratings for Chinese firms show high divergence (low convergence). In this regard, we suggest the following:

Firstly, rating agencies should provide as many additional details of the rating process as possible [19,20], such as more details about the underlying metrics and scoring processes, to make comparing and understanding rating information more convenient for information users.

Secondly, information users should use and understand ESG rating information from a more comprehensive perspective [21]. For example, in academic research, more ratings should be used in empirical analyses to improve the reliability and robustness of the conclusions.

Thirdly, market regulators should also promote the development of a sustainable corporate reporting system. The current Chinese reporting system does not force firms to report general hard information (e.g., carbon emissions), and allows them to use vague information for whitewashing purposes, which is one reason for the low convergence. In addition, market regulators should strengthen their regulation of the rating agencies [22].

Finally, we encourage third-party agencies other than these rating agencies to conduct comparative analyses and re-evaluations of these ratings to improve the efficiency of the market's use of multifaceted sustainability rating information.

## 5. Conclusions

As the world's largest developing market, China's sustainability transformation has led to a dramatic increase in the demand for corporate sustainability information and, as a result, the rapid growth of ESG rating businesses. This variety brings substantial references to the market, but may also create confusion for information users. Do the different ESG rating results converge? How great is this convergence (divergence)? To address the above issues, we selected seven ESG ratings from the international ratings firms Bloomberg, MSCI, and FTSE, and from the domestic ratings firms SynTao, RKS, SSII, and CASVI to carry out the analyses. First, we compared the differences between the rating systems, and then used the Hushen 300 index firms in 2019 as samples to carry out ranking overlap and rating correlation analysis. The analysis results showed the following: (1) Among the 195 sample firms covered by all seven ratings, the overlap degree of the top 100 rankings of each rating is low. For example, only approximately one third of the firms (35 firms) are recognized in the top 100 of all seven ratings. (2) The correlation among the ratings is also low, the correlation between two ratings is mostly below 0.7, and the average correlation is only 0.411. The above results show that due to the individually designed rating systems and the lack of mutual communication in the rating process, the results of different ESG ratings for Chinese firms show a high degree of divergence (low convergence).

Future research could include the following. Firstly, the SRI market has developed further given the current policy situation of the Chinese government (e.g., "double carbon policy"), and the ESG ratings business in China has continued to grow. For example, SSII has begun to update its ESG rating system, international agencies such as Bloomberg have continued to increase their coverage of Chinese firms, and large domestic securities firms such as the China Securities Index (CSI) have begun to launch ESG rating services. This paper encourages researchers to track the convergence (divergence) of ESG ratings based on China's post pandemic period (after 2022). Secondly, further research is needed to explore the factors that affect the ESG rating convergence (divergence) of Chinese firms and whether divergent rating results affect market activities (e.g., influence analysts' reports).

**Author Contributions:** M.Z.: conceptualization, design, manuscript preparation, and supervised this project. Y.Z. and H.Y.: data preparation and design, formal analysis, original draft preparation, and manuscript preparation sections. Each author contributed to the conceptualization and writing of this paper. All authors have read and agreed to the published version of the manuscript.

**Funding:** This research is funded by the National Natural Science Foundation of China (Grant number 71902090).

**Institutional Review Board Statement:** Not applicable.

**Informed Consent Statement:** Not applicable.

**Data Availability Statement:** Not applicable.

**Conflicts of Interest:** The authors declare no conflict of interest.

**Appendix A**

Based on the main databases of accounting and business and management publications (e.g., Google Scholar and Web of Sciences), we selected papers according to the following criteria: (1) the title, abstract and keywords of the paper contain the following terms: "Environmental, Social and Governance" (or "ESG") and "China"; (2) the paper preferably be a publication on a journal with important influence, e.g., journals included in ABS 2021 or in JCR top 50%; (3) the time span selected is from 2019 to 2023; (4) it must be an empirical paper, and the research object is Chinese A-share firms or related to the Chinese A-share market; (5) at least one of the ESG ratings (proxy ESG/sustainable performance or disclosure) of Bloomberg, FTSE, MSCI, SynTao, RKS, SSII, and CASVI are used for the main regression or robustness test in the study. After the final screening, we obtained 41 studies, and the content of these studies is summarized in Table A1. For example, we searched "ESG and China" on Google Scholar with the time limit of 2019–2023, and filtered out a paper "ESG and Firm's Default Risk" published in "Finance Research Letters"; next, by reading the research design part of the literature, we confirmed that it took Chinese listed firms as research samples, and used one of the seven ratings for empirical analysis (the sample of this literature was Chinese listed firms from 2015 to 2020, and the SSII ESG rating was used in the empirical test); therefore, we marked and summarized this paper.

**Table A1.** Overview of historical studies using ESG ratings.

| Study | Independent Variable | Dependent Variable | ESG Measure | Sampling Period | Obs |
|---|---|---|---|---|---|
| Chang, Cheng, Wang, Liu, and Hu (2023) [23] | ESG performance (+) | Corporate financing efficiency | SSII | 2013–2019 | 2100 |
| Li et al. (2023) [24] | ESG performance (+) | Corporate innovation | SSII (main test), Hexun and Bloomberg (robustness test) | 2009–2020 | 28,636 |
| Feng, Goodell, and Shen (2022) [25] | ESG ratings (-) | Stock price crash risk | SSII | 2009–2020 | 24,193 |

**Table A1.** *Cont.*

| Study | Independent Variable | Dependent Variable | ESG Measure | Sampling Period | Obs |
|---|---|---|---|---|---|
| Li, Zhang, and Zhao (2022) [26] | ESG performance (-) | Firms' default risk | SSII | 2015–2020 | 185,125 |
| Deng, Li, and Ren (2023) [27] | ESG performance (+) | Total factor productivity (TFP) | SSII (main test) and Bloomberg (robustness test) | 2016–2020 | 11,544 |
| Hu, Zou, and Yin (2023) [28] | ESG performance (+) | Stock price synchronicity | SSII | 2010–2021 | 24,544 |
| Kong (2023) [29] | ESG performance (-) | The cost of debt financing | SSII | 2009–2021 | 16,833 |
| Luo, Wei, and He (2023) [30] | ESG performance (+) | The access to trade credit | SSII (main test) and Hexun (robustness test) | 2011–2019 | 14,201 |
| Lian, Ye, Zhang, and Zhang (2023) [31] | ESG performance (-) | Bond credit spreads | SSII | 2009–2020 | 12,153 |
| Chen, Li, Zeng, and Zhu (2023) [32] | ESG performance (+) | The cost of equity capital | SSII (main test) and Bloomberg (robustness test) | 2010–2020 | 15,633 (main test) 7585 (robustness test) |
| Li, Lian, and Xu (2023) [33] | ESG performance (+) | Peer firms' green innovation | SSII (main test) Bloomberg (robustness test) | 2012–2020 | 11,259 (main test) 5964 (robustness test) |
| Zeng and Jiang (2023) [34] | ESG performance (+) | Corporate performances | SSII | 2009–2021 | 1247 |
| Tian and Tian (2022) [15] | ESG performance (+) | Corporate trade credit financing | SSII (main test) and Bloomberg (robustness test) | 2009–2020 | 32,534 (main test) 10,188 (robustness test) |
| Zhou, Liu and Luo (2022) [35] | ESG performance (+) | Market value of the firm | SynTao | 2014–2019 | 1002 |
| Broadstock, Chan, Cheng, and Wang (2021) [36] | ESG performance (+) | Financial risk during financial crisis | SynTao | 2015–2020 | 300 |
| Wang, Ma, Dong, and Zhang (2023) [37] | ESG rating (+) | Corporate green innovation | SynTao (main test) and CASVI (robustness test) | 2013–2019 | 18,790 |
| Deng and Cheng (2019) [38] | ESG indices (+) | Stock market performance | SynTao (main test) and CASVI (robustness test) | 2013–2019 | 5864 (main test) 3763 (robustness test) |
| Yang, Du, Zhang, Tong, and Zhou (2021) [39] | ESG disclosure (+) | Corporate bond credit spreads | SynTao | 2015–2020 | 2103 |
| Xu, Liu and Shang (2020) [40] | ESG performance (+) | Green invention | SynTao | 2015–2018 | 739 |
| He, Du, and Yu (2022) [41] | ESG performance (-) | Manager misconduct | Hexun (main test) and RKS (robustness test) | 2010–2020 | 23,741 (main test) 6030 (robustness test) |
| Yu, Liu, Cheng, and Lee (2022) [42] | ESG scores (no link) | Stock returns | RKS | 2020–2021 | 300 |
| Jun, Shiyong, and Yi (2022) [43] | ESG disclosure (non-linear) | Intangible capital | WIND and FTSE | 2017–2020 | 4593 |
| Li, Yin, and Liu (2022) [5] | ESG rating (-) | Stock price crash risk | Bloomberg, FTSE, MSCI, SSII, STGE, and CASVI | 2016–2020 | 13,919 (Only the CASVI sample is 11,765) |
| Zhang, Zhao, and He (2022) [44] | ESG performance (non-linear) | Portfolio excess returns | Bloomberg | 2005–2018 | 10,808 |
| Chen and Xie (2022) [45] | ESG disclosure (+) | Corporate financial performance | Bloomberg | 2011–2020 | 11,382 |
| Yuan, Li, Xu, and Shang (2022) [46] | ESG disclosure (-) | Corporate financial irregularity risks | Bloomberg (main test) and RKS (robustness test) | 2010–2020 | 7123 (main test) 4490 (robustness test) |
| Chen, Khurram, Gao, Abedin, and Lucey (2023) [47] | ESG disclosure (+) | Technological innovation capability | Bloomberg | 2011–2019 | 7146 |
| Ge, Xiao, Li, and Dai (2022) [48] | ESG performance (+) | High-quality development of firms | Bloomberg | 2011–2019 | 4985 |
| Liu, Zhu, Yang, and Chu (2022) [49] | ESG configuration | Corporate financial performance | Bloomberg | 2016–2020 | 210 |
| Zhou and Zhou (2022) [50] | ESG performance (-) | The stock price volatility of firms | MSCI | 2019–2020 | 1021 |
| Cheng, Lee, Li, and Tsang (2023) [51] | ESG proportion (+) and pillar mix efficiency heterogeneity (non-linear) | Financial performance | MSCI | 2015–2019 | 1108 |

**Table A1.** *Cont.*

| Study | Independent Variable | Dependent Variable | ESG Measure | Sampling Period | Obs |
|---|---|---|---|---|---|
| Baker, Boulton, Braga-Alves, and Morey (2021) [52] | ESG government ratings (-) | Firm-level IPO underpricing | MSCI | 2008–2018 | 7446 |
| Mu, Liu, Tao, and Ye (2023) [53] | Digital finance (+) | ESG performance | SSII | 2011–2020 | 26,294 |
| Wang, Lin, Fu, and Chen (2023) [54] | Institutional ownership heterogeneity (+) | ESG performance | SSII (main test) and Refinitiv (robustness test) | 2012–2021 | 23,607 (main test) 1124 (robustness test) |
| Wang, Peng, Tang, and Wu (2022) [55] | The development of fintech (+) | ESG performance | SSII (main test) and CASVI (robustness test) | 2010–2020 | 10,241 |
| Liu, Xiong, Gao, and Zhang (2022) [56] | Institutional shareholdings Heterogeneity (+) | ESG performance | Bloomberg (main test) and MSCI (robustness test) | 2010–2020 | 8421 |
| Meng and Zhu (2023) [57] | Female executives (+) | ESG performance | Bloomberg | 2011–2020 | 10,123 |
| Fang, Nie, and Shen (2023) [58] | Enterprise digitization (+) | ESG performance | Bloomberg (main test), SSII and Hexun (robustness test) | 2012–2020 | 2276 (main test) 2753 and 2036 (robustness test) |
| Yang, Guo, and Fan (2023) [59] | The network centrality of institutional investors (+) | ESG performance | Bloomberg (main test) and SynTao (robustness test) | 2009–2020 | 7562 (main test) 2876 (robustness test) |
| Wang, Sun, Wang, Hua, and Wu (2023) [60] | Environmental uncertainty (-) | ESG performance | Bloomberg (main test) and SSII (robustness test) | 2011–2020 | 5570 |
| Ren, Zeng, and Zhao (2023) [61] | Digital finance (+) | ESG performance | Bloomberg (main test) and SSII (robustness test) | 2011–2020 | 7223 (main test) 6798 (robustness test) |

**Appendix B**

In this section, we investigated the differences in regression analyses using different ratings. We regressed each rating using five variables mostly used in the relevant historical literature: (1) firm size (SIZE), measured as the natural log of total assets; (2) firm age (AGE), measured as the natural log of the number of years since the firm was listed; (3) financial leverage (LEV), measured as the liabilities/assets; (4) financial performance (ROE), measured as the return on equity; (5) shareholder structure (Balance), measured as the sum of the shareholdings proportion of the second- to the tenth-largest shareholder/the shareholding proportion of the largest shareholder. We designed Model (A1) as follows:

$$\text{ESG Rating}_i = \alpha_0 + \alpha_i \text{SIZE}_i + \alpha_2 \text{AGE}_i + \alpha_3 \text{LEV}_i + \alpha_4 \text{ROE}_i + \alpha_5 \text{Balance}_i + \varepsilon_i \quad \text{(A1)}$$

We used 195 firms that are jointly covered by all seven agencies as our regression sample (consistent with Section 3). Due to different financial reporting rules, financial firms were excluded, and the observations dropped to 157. To eliminate the effect of outliers, we winsorized all continuous variables at the 1st and 99th percentiles.

The regression results are presented in Table A2. In columns (1) to (7), the dependent variables are 0–1 standardized ESG scores from Bloomberg, FTSE, MSCI, SynTao, RKS, SSII, and CASVI, respectively. Firstly, except for SynTao, SIZE has a positive effect on ESG scores from all the other ratings ($p < 0.05$), and the significant coefficient ranges from 0.036 to 0.083. This result indicates that larger firms have better ESG ratings. Except for SSII and CASVI (They are grade ratings), the regression coefficients of the AGE are significantly negative ($p < 0.10$), and the significant coefficient ranges from $-0.068$ to $-0.092$. This suggests that firms that are listed later have better ESG ratings. The regression coefficient of LEV is negative and significant only in the regression using CASVI ($\alpha = -0.204$, $p < 0.05$); the rest are not significant. The coefficients on ROE vary considerably across ratings, and are only significant in regressions using Bloomberg, FTSE, and SynTao ($\alpha = -0.370$,

$p < 0.10$; $\alpha = -0.476$, $p < 0.05$; $\alpha = -0.414$, $p < 0.05$). There is also a large difference in the coefficients of Balance, which are only significant when using Bloomberg, FTSE, MSCI, and RKS ($\alpha = 3.563$, $p < 0.10$; $\alpha = 4.825$, $p < 0.05$; $\alpha = 3.687$, $p < 0.10$; $\alpha = 2.947$, $p < 0.10$). The above results show large variations in the regression results using different ratings, which indicates that there is a high divergence (low convergence) among these ratings. This is consistent with our previous analysis conclusions. Comparatively, the regression results of Bloomberg and FTSE have the highest consistency, and the regression coefficients of the two are basically consistent in the five variables. The results of MSCI are also consistent with the two, except for the ROE. In general, the results of the three international ratings in this regression analysis are higher than those of the domestic ratings. Among the domestic ratings, the results of RKS have the highest consistency with those of the three international ratings, and all of the coefficients are similar to those of the analysis of the international ratings, except for the ROE. However, SSII and CASVI ratings have the lowest consistency, and they use the grade rating system. They differ in the coefficients of all variables except for SIZE.

**Table A2.** Comparative regression analysis.

|  | (1) Bloomberg | (2) FTSE | (3) MSCI | (4) SynTao | (5) RKS | (6) SSII | (7) CASVI |
|---|---|---|---|---|---|---|---|
| SIZE | **0.075 \*\*\*** | **0.036 \*\*** | **0.081 \*\*\*** | 0.015 | **0.066 \*\*\*** | **0.054 \*\*\*** | **0.083 \*\*\*** |
|  | (4.873) | (2.585) | (5.442) | (0.995) | (5.596) | (3.184) | (6.958) |
| AGE | **−0.090 \*\*** | **−0.089 \*\*\*** | **−0.086 \*\*** | **−0.068 \*** | **−0.092 \*\*\*** | 0.019 | −0.004 |
|  | (−2.459) | (−2.671) | (−2.406) | (−1.898) | (−3.274) | (0.470) | (−0.150) |
| LEV | −0.145 | 0.038 | −0.158 | −0.117 | −0.121 | −0.194 | **−0.204 \*\*** |
|  | (−1.320) | (0.387) | (−1.489) | (−1.098) | (−1.443) | (−1.607) | (−2.411) |
| ROE | **−0.370 \*** | **−0.476 \*\*** | **−0.053** | **−0.414 \*\*** | −0.053 | −0.173 | 0.032 |
|  | (−1.795) | (−2.546) | (−0.265) | (−2.075) | (−0.334) | (−0.759) | (0.203) |
| Balance | **3.563 \*** | **4.825 \*\*** | **3.687 \*** | 2.192 | **2.947 \*** | 1.614 | 2.169 |
|  | (1.720) | (2.564) | (1.841) | (1.091) | (1.856) | (0.706) | (1.355) |
| _cons | −1.167 \*\*\* | −0.275 | −1.321 \*\*\* | 0.283 | −1.074 \*\*\* | −0.681 | −1.334 \*\*\* |
|  | (−3.127) | (−0.812) | (−3.661) | (0.783) | (−3.753) | (−1.651) | (−4.624) |
| N | 157 | 157 | 157 | 157 | 157 | 157 | 157 |
| r2 | 0.222 | 0.197 | 0.224 | 0.063 | 0.252 | 0.078 | 0.271 |

Note: Bolded numbers are significant coefficients; *t* statistics are reported in parentheses; * $p < 0.1$, ** $p < 0.05$, *** $p < 0.01$.

**Appendix C**

Table A3 shows the results of the Spearman correlation analysis of the ESG rating scores. In general, the correlation coefficients between the two ratings range from 0.116 (FTSE and SSII) to 0.719 (Bloomberg and SynTao), with an average of 0.450. Based on each rating, Bloomberg has the highest correlation with other ratings, with an average correlation coefficient of 0.570; the other results (MSCI, RKS, SynTao, FTSE) are also similar, with mean correlation coefficients of around 0.5; SSII and CASVI remain lower correlation coefficients with other ratings, with mean coefficients of 0.248 and 0.348, respectively. Apparently, the Spearman correlation analysis presents consistent results with Section 3.5.

**Table A3.** Spearman correlation coefficients between ESG ratings.

|  | **Bloomberg** | **FTSE** | **MSCI** | **SynTao** | **RKS** | **SSII** | **CASVI** |
|---|---|---|---|---|---|---|---|
| Bloomberg | 1 |  |  |  |  |  |  |
| FTSE | 0.608 \*\*\* | 1 |  |  |  |  |  |
| MSCI | 0.670 \*\*\* | 0.582 \*\*\* | 1 |  |  |  |  |
| SynTao | 0.719 \*\*\* | 0.607 \*\*\* | 0.571 \*\*\* | 1 |  |  |  |
| RKS | 0.685 \*\*\* | 0.670 \*\*\* | 0.629 \*\*\* | 0.573 \*\*\* | 1 |  |  |
| SSII | 0.295 \*\*\* | 0.116 | 0.227 \*\*\* | 0.257 \*\*\* | 0.154 \*\* | 1 |  |
| CASVI | 0.441 \*\*\* | 0.203 \*\*\* | 0.397 \*\*\* | 0.275 \*\*\* | 0.331 \*\*\* | 0.438 \*\*\* | 1 |
| Mean of coefficients | 0.570 | 0.464 | 0.513 | 0.500 | 0.507 | 0.248 | 0.348 |

Note: ** $p < 0.05$, *** $p < 0.01$.

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
