# Peer review of "Do ESG Ratings of Chinese Firms Converge or Diverge? A Comparative Analysis Based on Multiple Domestic and International Ratings"

_sustainability, doi:10.3390/su151612573_

Round 1
Reviewer 1 Report
Thank you for the opportunity to review such an interesting paper!
Just a few considerations authors should reflect on:
- the methodology from section 3 Convergence analysis of ESG ratings of Chinese firms is not corroborated at all with the existing literature (there is no mention of other studies that did the convergence analysis of other ratings). It is necessary to document the paper from the literature also from this perspective.
-a broader discussion about the reasons for the differences between ratings is necessary (the authors send it elegantly to future research directions...)
-after the analysis, can you say if some of the ratings are better, more complete than others? Is it possible to rank them, possibly, separately domestic and international ratings?
- the authors fear that “If ESG ratings diverge too much, the reliability and robustness of the conclusions of these studies may be in doubt”. However, from the list in Annex a1 it appears that the studies that used at least 2 ratings use the 2nd one for robustness check (which we can assume confirms the results of the reference models), which does not justify the authors' fear. How do you comment on this?
Congratulations on this paper!
Author Response
We are grateful for your careful reading of our paper and more importantly, your positive and constructive comments. We have incorporated all your comments and suggestions into this version of the manuscript. Because of your thoughtful review of the manuscript, we believe that our revision has greatly improved the paper. We hope that we have succeeded in addressing the concerns raised in your comments.
Below, we detail our point-by-point responses to your comments (in bold and italic fonts). Our response starts with ***.
Reviewer-1 Comments:
Comments to the Author
Thank you for the opportunity to review such an interesting paper!
Just a few considerations authors should reflect on:
- the methodology from section 3 Convergence analysis of ESG ratings of Chinese firms is not corroborated at all with the existing literature (there is no mention of other studies that did the convergence analysis of other ratings). It is necessary to document the paper from the literature also from this perspective.
*** We are very grateful to you for these positive comments on the academic and practical relevance of our paper. we have added the literature basis for our choice of methods in the added Section 3.1.
- a broader discussion about the reasons for the differences between ratings is necessary (the authors send it elegantly to future research directions...)
*** Thanks for the tip. We have mentioned some possible reasons in the main text, for example, in the last sentence of the first paragraph of Section 5: "… due to the individually designed rating systems and the lack of mutual communication in the rating process…”. However, the discussion on the causes of divergence should be a large research topic. For example, (Christensen, Serafeim, and Sikochi, 2022) have a paper specifically on the determinants of ESG divergence in developed markets. The full paper is 64 pages long. It is difficult to make a complete discussion in the existing paper, but we add some relevant thoughts in the conclusion (future research directions).
Reference:
Christensen, D. M.Serafeim, G., &Sikochi, A. (2022). Why is Corporate Virtue in the Eye of The Beholder? The Case of ESG Ratings. The Accounting review, 97(1), 147-175. 10.2308/TAR-2019-0506
- after the analysis, can you say if some of the ratings are better, more complete than others? Is it possible to rank them, possibly, separately domestic and international ratings?
*** Thanks to the concern. Since these ratings agencies do not disclose all the details of the ratings (international ratings disclose more information, but again not all the details of the ratings), it is difficult for us to make convincing judgments about the strengths and weaknesses of the ratings (moreover, if we provide a ranking, it may lead to further disputes). But we have added some related implicit inferences, for example, at the end of the paragraph in Section 3.3, based on the 0-1 standardized ratings, we argue that “The above results indicate that the scoring results of different ratings have different tendencies: relatively, the ratings of Chinese firms by international agencies are more conservative; among domestic ratings, established sustainability consultants SynTao and RKS also report more conservative results, but financial index firm SSII and nongovernmental organization CASVI are more optimistic.”
- the authors fear that “If ESG ratings diverge too much, the reliability and robustness of the conclusions of these studies may be in doubt”. However, from the list in Annex a1 it appears that the studies that used at least 2 ratings use the 2nd one for robustness check (which we can assume confirms the results of the reference models), which does not justify the authors' fear. How do you comment on this?
*** We thank the reviewer for the suggestions. In this regard, we believe that it is still difficult to ensure the robustness of the conclusions even when two ratings are used; for example, in our analysis, we have found that the top 100 common coverage between pairs of two ratings ranges from 66.36% to 82.35%, but the overlap rate between the three international ratings and the four domestic ratings falls to 57% and 46%, respectively, and that the common overlap rate between seven ratings falls to 35%, so even if the conclusions obtained using two ratings are consistent, the robustness of the conclusions when using a third or more ratings is still questionable.

Reviewer 2 Report
· 1. Avoid using question in the title, add research method in the title.
The authors should use full name of ESG in the abstract part for the first time of mentioning this word.
2. The authors may consider simplifying the content of the abstract part. State academic, practical and policy contributions, research gap that has been filled and originality of this paper.
3. The authors may need to consider revising the content of the
introduction part, it is not necessary to reveal the results of this paper
in this stage.
4. Research questions should be stated in introduction
5. The section 2 may be a part of the literature review part, other research
background may also be added into this part. 2.1.1, missing citation for the whole section. ESG literature should be added (2022), Modularity clustering of economic development and ESG attributes in prefabricated building research. Front. Environ. Sci. 10:977887. doi: 10.3389/fenvs.2022.977887
6. The section 3 may be a part of research methods, and a section with this title should be necessary.
7. “Table 6 reports descriptive statistics, including observation, mean, standard deviation, median, and so on, of ESG rating score based on the 195 sample firms” the authors may slightly revise the words of this sentence. 3. Convergence analysis, what is that? Please state at the beginning with the statistics models if applicable.
8. Colored figures should be used and add citations where applicable. The authors may need to add a brief introduction of the Pearson correlation https://www.frontiersin.org/articles/10.3389/fpubh.2022.993700/full
· correlation analysis with reference.
9. A discussion part should be added based on the results obtained from this research.
10.Some of the references may seem a bit outdated, the authors may
consider revising them.
11. Move footnote to main text.
12. Polish English.
13. More complex methods in analysis should be added on top of correlation.
NA
Author Response
We are grateful for your careful reading of our paper and more importantly, your positive and constructive comments. We have incorporated all your comments and suggestions into this version of the manuscript. Because of your thoughtful review of the manuscript, we believe that our revision has greatly improved the paper. We hope that we have succeeded in addressing the concerns raised in your comments.
Below, we detail our point-by-point responses to your comments (in bold and italic fonts). Our response starts with ***.
Reviewer-2 Comments:
Comments to the Author
- Avoid using question in the title, add research method in the title.
The authors should use full name of ESG in the abstract part for the first time of mentioning this word.
*** Sincerely thank you for your comments. First of all, this title is used in this paper to pay homage to the classic literature (Chatterji, Durand, Levine, and Touboul, 2016) (published on the top journal AMJ, 736 citations on Google Scholar). Moreover, listing the research question as a title is a common proposition in academic areas of management and economics, e.g., (Bae, El Ghoul, Gong, and Guedhami, 2021; Burke, 2022; Chan, Jiang, Wu, Xu, and Zeng, 2020; Guo, Zha, Lee, and Tang, 2020; Safiullah, Alam, and Islam, 2022). Thus, we argue that this title is appropriate and may increase the visibility and citation of the study.
Reference:
Bae, K.El Ghoul, S.Gong, Z., &Guedhami, O. (2021). Does CSR matter in times of crisis? Evidence from the COVID-19 pandemic. Journal of Corporate Finance, 67(101876. 10.1016/j.jcorpfin.2020.101876
Burke, J. J. (2022). Do Boards Take Environmental, Social, and Governance Issues Seriously? Evidence from Media Coverage and CEO Dismissals. Journal of Business Ethics, 176(4), 647-671. 10.1007/s10551-020-04715-x
Chan, K. C.Jiang, X.Wu, D.Xu, N., &Zeng, H. (2020). When Is the Client King? Evidence from Affiliated-Analyst Recommendations in China's Split-Share Reform. Contemporary Accounting Research, 37(2), 1044-1072. https://doi.org/10.1111/1911-3846.12550
Chatterji, A. K.Durand, R.Levine, D. I., &Touboul, S. (2016). Do ratings of firms converge? Implications for managers, investors and strategy researchers. Strategic Management Journal, 37(8), 1597-1614. 10.1002/smj.2407
Guo, T.Zha, G.Lee, C. L., &Tang, Q. (2020). Does corporate green ranking reflect carbon-mitigation performance? Journal of Cleaner Production, 277(123601. 10.1016/j.jclepro.2020.123601
Safiullah, M.Alam, M., &Islam, M. (2022). Do all institutional investors care about corporate carbon emissions? Energy Economics, 106376. 10.1016/j.eneco.2022.106376
- The authors may consider simplifying the content of the abstract part. State academic, practical and policy contributions, research gap that has been filled and originality of this paper.
*** Thanks to the comments, we have made further adjustments, reducing the original 670 words to 563 and reorganizing our description of academic, practical and policy contributions and so on.
- The authors may need to consider revising the content of the introduction part, it is not necessary to reveal the results of this paper in this stage.
*** Thanks for the suggestion, we made adjustments. Please read the new introduction.
- Research questions should be stated in introduction.
*** We appreciate the correction, and we have added "Therefore, we raise the research questions: do significant divergences exist? To what extent do these ESG rating results converge or diverge? ..." (line 51) before the research question in the introduction section.
- The section 2 may be a part of the literature review part, other research background may also be added into this part. 2.1.1, missing citation for the whole section. ESG literature should be added (2022), Modularity clustering of economic development and ESG attributes in prefabricated building research. Front. Environ. Sci. 10:977887. doi: 10.3389/fenvs.2022.977887
*** Thanks for the correction, we have added this reference in our study. Please read the references (No.3).
- The section 3 may be a part of research methods, and a section with this title should be necessary.
*** Based on the reviewer's comments, we have added a new subsection "3.1. Analysis design" to the first paragraph under Section 3. Please read Section 3 in main text.
- “Table 6 reports descriptive statistics, including observation, mean, standard deviation, median, and so on, of ESG rating score based on the 195 sample firms” the authors may slightly revise the words of this sentence. 3. Convergence analysis, what is that? Please state at the beginning with the statistics models if applicable.
*** Thanks for the suggestion. To avoid misunderstanding, the corresponding sublabel "3. Convergence analysis of ESG ratings of Chinese firms" has been revised to "Analysis for the convergence/divergence of Chinese ESG ratings". We use the overlap analysis of the list and the correlation analysis of the ratings scores to illustrate the degree of convergence/divergence of the ratings.
- Colored figures should be used and add citations where applicable. The authors may need to add a brief introduction of the Pearson correlation https://www.frontiersin.org/articles/10.3389/fpubh.2022.993700/full correlation analysis with reference.
*** The reviewer's comments are very meaningful. We have changed the color illustrations and added this reference. Please read the references (No.17).
- A discussion part should be added based on the results obtained from this research.
*** Thanks for the tip. The original Conclusion section has been reworded to read in several parts, namely, “4. Discussion” and “5. Conclusion”.
- Some of the references may seem a bit outdated, the authors may consider revising them.
*** The old references are necessary and they are used to supplement the description of the research background, for example, the literature "Urban consumers' attitudes towards the safety of milk powder after the melamine scandal in 2008 and the factors influencing the attitudes"(Zhou and Wang, 2011) is for the description of the Sanlu incident, therefore, we feel that it is necessary to keep these old references.
Reference:
Zhou, Y., &Wang, E. (2011). Urban consumers' attitudes towards the safety of milk powder after the melamine scandal in 2008 and the factors influencing the attitudes. China Agricultural Economic Review, 3(1), 101-111. 10.1108/17561371111103589
- Move footnote to main text.
*** Thanks for the reviewer's reminder, we have moved all footnotes to main text.
- Polish English.
*** We apologize for the misunderstanding. We have polished this paper, please see the language certification in the attachment.
- More complex methods in analysis should be added on top of correlation.
*** The methodology used in this study refers to several classic literatures (Berg, Kölbel, and Rigobon, 2022; Chatterji, Durand, Levine, and Touboul, 2016; Christensen, Serafeim, and Sikochi, 2022; Dimson, Marsh, and Staunton, 2020; Gibson Brandon, Krueger, and Schmidt, 2021), these two methods are intuitive and easy to understand, especially for market practitioners to fully understand the divergence of these rating results (considering that the purpose of this study is to help market participants better understand the analysis of ESG ratings, so we worry that an overly complex approach may be detrimental to their understanding). Of course, if the reviewer thinks that any complex method is more accurate and specific, please point out, and we can try to use it.
Reference:
Berg, F.Kölbel, J. F., &Rigobon, R. (2022). Aggregate Confusion: The Divergence of ESG Ratings. Review of Finance, 26(6), 1315-1344. 10.1093/rof/rfac033
Chatterji, A. K.Durand, R.Levine, D. I., &Touboul, S. (2016). Do ratings of firms converge? Implications for managers, investors and strategy researchers. Strategic Management Journal, 37(8), 1597-1614. 10.1002/smj.2407
Christensen, D. M.Serafeim, G., &Sikochi, A. (2022). Why is Corporate Virtue in the Eye of The Beholder? The Case of ESG Ratings. The Accounting review, 97(1), 147-175. 10.2308/TAR-2019-0506
Dimson, E.Marsh, P., &Staunton, M. (2020). Divergent ESG Ratings. Journal of portfolio management, 47(1), 75-87. 10.3905/JPM.2020.1.175
Gibson Brandon, R.Krueger, P., &Schmidt, P. S. (2021). ESG Rating Disagreement and Stock Returns. Financial analysts journal, 77(4), 104-127. 10.1080/0015198X.2021.1963186

Reviewer 3 Report
The paper investigates an intriguing and realistic topic. This paper first compares the differences of seven major ESG ratings in the Chinese market in terms of metric system, information source and so on, and then analyzes the differences of these rating results by taking the 2019 Hushen 300 index firms as a sample. Evidences revealed in the paper suggest that there is a high divergence in these ratings, and these results may provide useful references for both academic researchers and market participants. The paper still has some details that could be improved:
(1) Why did the paper choose these seven ratings? The authors can add some description as appropriate.
(2) The paper chose to use 49 articles in its example of ESG rating’s application, but the process of selecting these articles could be supplemented with a more detailed description.
(3) The paper also has some typos, for example, in the last column of row 4 of Table 1, "Covering 5% of A-share listed firms in 2018, the proportion will be raised to 20% in 2019."; on the left side of Figure 1, "Ratio of ESG Ratings" is used, which seems to be inconsistent with the title of the figure." The authors need to correct these minor typos.
I suggest that the authors invite a professional language editor to polish the full text.
Author Response
We are grateful for your careful reading of our paper and more importantly, your positive and constructive comments. We have incorporated all your comments and suggestions into this version of the manuscript. Because of your thoughtful review of the manuscript, we believe that our revision has greatly improved the paper. We hope that we have succeeded in addressing the concerns raised in your comments.
Below, we detail our point-by-point responses to your comments (in bold and italic fonts). Our response starts with ***.
Reviewer-3 Comments:
Comments to the Author
The paper investigates an intriguing and realistic topic. This paper first compares the differences of seven major ESG ratings in the Chinese market in terms of metric system, information source and so on, and then analyzes the differences of these rating results by taking the 2019 Hushen 300 index firms as a sample. Evidences revealed in the paper suggest that there is a high divergence in these ratings, and these results may provide useful references for both academic researchers and market participants. The paper still has some details that could be improved:
- Why did the paper choose these seven ratings? The authors can add some description as appropriate.
*** Thanks to the reviewer's comments, we have added clarifications in the introduction: “the above 7 ESG ratings are considered by some studies to have an important impact on the Chinese market, e.g., (Li, Yin, and Liu, 2022) and (Liu, 2022)” (line 55-56).
- The paper chose to use 49 articles in its example of ESG rating’s application, but the process of selecting these articles could be supplemented with a more detailed description.
*** We appreciate the reviewer's comments and we have added further clarifications to the relevant notes “Based on the main databases of accounting, business and management publications (e.g., Google Scholar and Web of Sciences), we select papers according to the following criteria: (1) The title, abstract and keywords of the paper contain the following terms: “Environmental, Social and Governance” (or “ESG”) and “China”; (2) It preferably be a publication on a journal with important influence, e.g., journals included in ABS 2021 or in JCR top 50%; (3) The time span is selected from 2019 to 2023; (4) It must be an empirical paper, and the research object is Chinese A-share firms or related to the Chinese A-share market; (5) At least one of the ESG ratings (proxy ESG/sustainable performance or disclosure) of Bloomberg, FTSE, MSCI, SynTao, RKS, SSII and CASVI are used for the main regression or robustness test in the study. After the final screening, we obtained 49 studies, and the content of these studies is summarized in Table a1. For example, we search "ESG and China" on Google Scholar with the time limit of 2019-2023, and filter out a paper " ESG and Firm's Default Risk " published in " Finance Research Letters "; next, by reading the research design part of the literature, we confirm that it takes Chinese listed firms as research samples and uses one of the seven ratings for empirical analysis (The sample of this literature is Chinese listed firms from 2015 to 2020, and SSII ESG rating is used in the empirical test.); therefore, we mark and summarize this paper” (Appendix A).
- The paper also has some typos, for example, in the last column of row 4 of Table 1, "Covering 5% of A-share listed firms in 2018, the proportion will be raised to 20% in 2019."; on the left side of Figure 1, "Ratio of ESG Ratings" is used, which seems to be inconsistent with the title of the figure." The authors need to correct these minor typos.
*** We thank the reviewer for their corrections, and we have carried out a more careful proofreading process.

Round 2
Reviewer 2 Report
Polish English.
The title has to change, not sure what you mean by diverge or converge. It is not in proper English.
There is no such thing as international or domestic. Readers come from everywhere. Domestic of an American will mean different from Chinese when we said what is domestic. I never read any journals from outside China claiming their research is domestic or international.
The title said domestic and international rating, Table 2 should state something about these? Which one is domestic and which one is international.
Discussion section should be lengthened with discussion based on previous publication.
Syntao. China sustainable investment review; Syntao: Bejing, 2022; p. The reference is incomplete.
Table a1. Related research statistics, what are the criteria of inclusion in independent variables? A lot of them may have 8 plus independent variables. The authors have to check.
Polish English.
Author Response
We would like to thank you for forwarding the reviewer’s comments to us. Below, we detail our point-by-point responses to your comments (in bold and italic fonts).
Our response starts with ***.
Comments:
Comments to the Author
1.The title has to change, not sure what you mean by diverge or converge. It is not in proper English.
*** Thanks for the reminder. However, we still insist on using this title, which references the following classic literature to increase the visibility and citation of this paper:
Chatterji, A., Durand, R., Levine, D. I., & Touboul, S. (2015). Do Ratings of Firms Converge? Implications for Managers, Investors and Strategy Researchers. Strategic Management Journal. (UTD24 journal)
Yang, M., Maresova, P., Akbar, A., Bento, P., & Liu, W. (2021). Convergence Or Disparity? A Cross-Country Analysis of Corporate Social Responsibility Reporting for Banking Industry in Nordic Countries and China. SAGE Open, 11(3), 1999480997.
Zhong, M., Xu, R., Liao, X., & Zhang, S. (2019). Do CSR Ratings Converge in China? A Comparison Between RKS and Hexun Scores. Sustainability, 11(14), 3921Chatterji, A.
- There is no such thing as international or domestic. Readers come from everywhere. Domestic of an American will mean different from Chinese when we said what is domestic. I never read any journals from outside China claiming their research is domestic or international.
*** Thanks for the comments. We have pointed out several times in the abstract (“Domestic agencies, such as SynTao …”, line 9-14) and the main text (“… we use 195 Hushen 300 index firms commonly covered by 7 domestic and international ESG ratings in 2019 …”, line 53-56) of the paper that the seven ratings we studied include three international ratings (Bloomberg, FTSE and MSCI), which cover not only Chinese firms but also firms in other countries. And four Chinese domestic ratings (SynTao, RKS, SSII, CASVI), which only cover Chinese firms. If the reviewer still has doubts, please read the following studies (Li, Yin, and Liu, 2022; Liu, 2022) or the methodology guidance of these ratings.
Li, S. Yin, P., &Liu, S. (2022). Evaluation of ESG Ratings for Chinese Listed Companies from the Perspective of Stock Price Crash Risk. Frontiers in Environmental Science, 10(10.3389/fenvs.2022.933639
Liu, M. (2022). Quantitative ESG disclosure and divergence of ESG ratings. Frontiers in Psychology, 13(10.3389/fpsyg.2022.936798
3.The title said domestic and international rating, Table 2 should state something about these? Which one is domestic and which one is international.
*** Please see the column (2) of Table 1, where we have indicated whether the seven rating categories are "domestic" or "international". Moreover, this information has also mentioned in the abstract and the introduction, we will not repeat it in Table 2.
- Discussion section should be lengthened with discussion based on previous publication.
*** Based on the reviewer's comments, we have added relevant previous literature in Discussion. Please read the discussion.
- China sustainable investment review; Syntao: Bejing, 2022; p 2. The reference is incomplete.
*** Thanks for the correction. This is a report from SynTao. We have changed it to the MDPI format, referring to https://dataexplorer.syntaogf.com/china-sustainable-investment-review-2022 for more details.
- Table a1. Related research statistics, what are the criteria of inclusion in independent variables? A lot of them may have 8 plus independent variables. The authors have to check.
*** Thanks for the suggestion. We have proofread Appendix A and found no problems. This table refers to the literature of some top journals, for example:
Gull, A.A.; Hussain, N.; Khan, S.A.; Khan, Z.; Saeed, A. Governing Corporate Social Responsibility Decoupling: The Effect of the Governance Committee on Corporate Social Responsibility Decoupling. J Bus Ethics 2022, doi:10.1007/s10551-022-05181-3.
Thank you again for your careful reading of the paper and for pointing out so many important issues. We have carefully revised our paper following your great comments.
Yours sincerely
Authors
